# A chip-scale atomic beam clock

Gabriela D. Martinez[1,2], Chao Li [3,4] ✉, Alexander Staron[1,2], John Kitching [1], Chandra Raman[3] & William R. McGehee [1] ✉

Atomic beams are a longstanding technology for atom-based sensors and clocks with widespread use in commercial frequency standards. Here, we report the demonstration of a chip-scale microwave atomic beam clock using coherent population trapping (CPT) interrogation in a passively pumped atomic beam device. The beam device consists of a hermetically sealed vacuum cell fabricated from an anodically bonded stack of glass and Si wafers in which lithographically defined capillaries produce Rb atomic beams and passive pumps maintain the vacuum environment. A prototype chip-scale clock is realized using Ramsey CPT spectroscopy of the atomic beam over a 10 mm distance and demonstrates a fractional frequency stability of $\approx 1.2 \times 10^{-9}/\sqrt{\tau}$ for integration times, τ, from 1 s to 250 s, limited by detection noise. Optimized atomic beam clocks based on this approach may exceed the long-term stability of existing chip-scale clocks, and leading long-term systematics are predicted to limit the ultimate fractional frequency stability below $10^{-12}$.

The development of low-power, chip-scale atomic devices including clocks and magnetometers has been enabled by advances in the optical interrogation of atoms confined in microfabricated vapor cells[1]. These miniaturized devices commonly use coherent population trapping (CPT) resonances in alkali atoms, which generate a coherent dark state between hyperfine atomic ground states using two optical fields in a Λ-scheme[2]. Optical probing of the microwave transition avoids the need for bulky microwave cavities, providing a compact and low-power method for probing the atoms and enabling battery-powered operation[3,4]. Buffer gases are commonly used to reduce the decoherence rate from wall collisions and narrow the atomic line. As a result, devices such as the chip-scale atomic clock (CSAC) can realize $\approx 10^{-11}$ fractional frequency stability at 1000 s of averaging while consuming only 120 mW of power[5]. Thermal drifts and aging of the buffer gas environment, along with light shifts and other systematics, contribute to the long-term instability of buffer gas systems and degrades clock performance in existing CSACs beyond 1000 s of averaging with a drift rate of $-10^{-9}$ per month.

Clocks based on atomic beams and laser-cooled gases operate in ultra-high vacuum (UHV) environments and avoid shifts from buffer gases, allowing for higher frequency stability and continuous averaging over periods of days or weeks. Laser-cooling technology underpins the most advanced atomic clocks[6], and while recent efforts in photonic integration[7,8] and vacuum technology[9–11] have advanced the state of the art, significant hurdles to miniaturization and low-power operation remain[12]. Atomic beams have played a significant role throughout the history of frequency metrology, serving as commercial frequency standards since the 1960s and as national frequency standards for realization of the SI second[13,14]. Miniaturized atomic beams[15–19] offer a path for exceeding the long-term stability of existing chip-scale devices while circumventing the complexity and power needs of more advanced laser-cooled schemes.

In this work, we demonstrate a chip-scale atomic beam clock built using a passively pumped Rb atomic beam device as shown in Fig. 1. The beam device contains a Rb reservoir that feeds a microcapillary array and generates Rb atomic beams in an internal, evacuated cavity. Fabrication of the device is realized using a stack of lithographically defined planar structures which are anodically bonded to form a hermetic package. Spectroscopic measurements of the atomic flux and beam collimation are presented to demonstrate the successful realization of the atomic beam device. The atom beam device presents a pathway for realizing low-power, low-drift atomic sensors using

[1]Time and Frequency Division, National Institute of Standards and Technology, Boulder, CO, USA. [2]Department of Physics, University of Colorado Boulder, Boulder, CO, USA. [3]School of Physics, Georgia Institute of Technology, Atlanta, GA, USA. [4]Present address: Research Laboratory of Electronics, Massachusetts Institute of Technology, Cambridge, MA, USA. ✉e-mail: lichao@gatech.edu; william.mcgehee@nist.gov

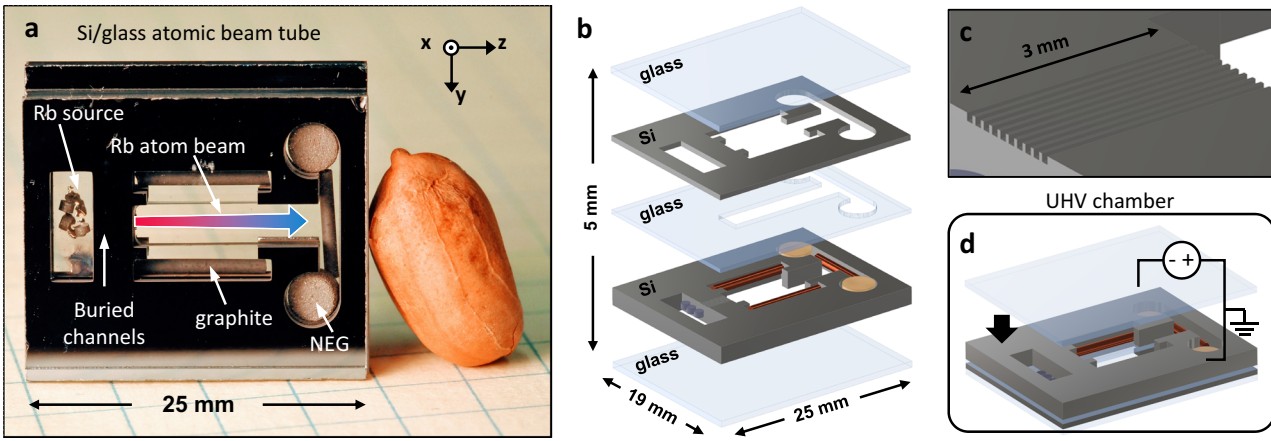

**Fig. 1 | Overview of chip-scale atomic beam device. a** Image of atomic beam device with labeled components (peanut for scale). Rb vapor in the source cavity feeds a buried microcapillary array and forms an atomic beam (indicated by a red-to-blue arrow) in the drift cavity. Non-evaporable getters (NEGs) and graphite maintain the vacuum environment in the device. **b** Expanded view of the beam device showing component layers as well as Rb pill dispensers, graphite rods, and NEG pumps. **c** Schematic of the microcapillary array etched in a Si wafer. Each capillary has a 100 μm × 100 μm square cross-section. The array collimates the atomic beam and provides differential pumping between the source and drift regions. **d** The final anodic bond which hermetically seals the device occurs in an ultra-high vacuum (UHV) chamber.

microfabricated components and supports further integration with advanced thermal and photonic packaging to realize highly manufacturable quantum sensors.

The microwave atomic beam clock is demonstrated using Ramsey CPT interrogation in the atomic beam device. Ramsey spectroscopy of the [87]Rb ground-state hyperfine transitions is measured across a 10 mm distance and demonstrates quantum coherence across the device. The magnetically insensitive $m_F = 0$ hyperfine transition is used to realize the atomic beam clock signal, and a clock short-term fractional frequency stability of $\approx 1.2 \times 10^{-9}$ is achieved at 1 s of integration in this prototype device. The performance of this beam clock is limited by technical noise, and an optimized cm-scale device is expected to achieve stability better than $10^{-10}$ at 1 s of integration with a stability floor below $10^{-12}$, supported by a detailed analysis of the sources of drift in atomic beam clocks.

## Results

The passively pumped atom beam device is fabricated from a multi-layer stack of Si and glass wafers as shown in Fig. 1. The layers are anodically bonded to form a hermetically sealed vacuum cell with dimensions of 25 mm × 23 mm × 5 mm and ≈0.4 cm³ of internal volume. Internal components include Rb molybdate Zr/Al pill-type dispensers for generating Rb vapor in an internal source cavity[20] as well as graphite and Zr/V/Fe non-evaporable getters (NEGs) in a separate drift cavity which maintain the vacuum environment. A series of microcapillaries connect the two internal cavities and produce atomic beams which freely propagate for 15 mm in the drift cavity. The device is heated to generate Rb vapor in the source cavity, and the atomic beam flux and divergence are defined by the capillary geometry[21]. Microfabrication allows for arbitrary modification of the shape, continuity, and divergence of the capillaries to control the atomic beam properties[18,19].

The arrangement of alternating glass and Si layers and internal components which comprise the beam device are shown in Fig. 1b. The features etched in Si are created using deep reactive ion etching (DRIE), and the cavity in the central glass layer is conventionally machined. The two transparent encapsulating glass layers are low-He-permeation aluminosilicate glass[22] with 700 μm thickness. The microcapillary array is etched into a 2 mm-thick Si layer which houses the internal components used for passive pumping and sourcing Rb. Two additional layers, a 600 μm-thick Si layer, and a 1 mm-thick borosilicate glass layer, act as a spacer to position the microcapillary array near the center of the device thickness and provide volume into which the atomic beam can expand in the drift cavity. The device is assembled by first anodically bonding the four upper layers under ambient conditions (see Methods) to create a preform structure. The four-layer preform is populated with the getters and Rb pill dispensers and topped with the final glass wafer. The stack is placed in an ultra-high vacuum (UHV) chamber and baked at 520 K for 20 h to degas the components, and the NEGs are thermally activated using laser heating to remove their passivation layer before sealing the device. The final interface is then anodically bonded to hermetically seal the vacuum device (see Fig. 1d).

Rb atomic beams are generated in the drift cavity as vapor from the source cavity flows through the microcapillary array[18]. The atomic flux is determined by the source region Rb density and the geometry of the capillary array, which consists of 10 straight capillaries with 100 μm × 100 μm square cross section, 50 μm spacing, and 3 mm length. The flux through the capillaries and angular profile of the atomic beam are well described by analytic molecular flow models based on the capillary's aspect ratio $L/a$, where L is the capillary length and a is its width[21]. The near-axis flux is similar to that of a "cosine" emitter for angles $\theta$ less than a/L from normal. The total flux through the capillary array $F_n = \frac{1}{4} w\, n_{Rb,1}\, \bar{v} A_c$, where $w = 1/(1 + 3L/4a)$ is the transmission probability of the channel, $n_{Rb,1}$ is the Rb density in the source region, $\bar{v}$ is the mean thermal speed of the atoms, and $A_c$ is the cross-sectional area of the capillary array (here $10a^2$). More complex capillary geometries such as the non-parallel or cascaded collimators can be used to further engineer the beam profile or reduce off-axis flux[18,19].

The performance of the atomic beam device is measured using optical spectroscopy on the Rb $D_2$ line at ~780 nm. The atoms are probed using a 5 μW elliptical laser beam with $w_y \approx 2100$ μm, $w_z \approx 350$ μm (1/e² radius) normally incident on the device surface (propagating along the x axis). The total density $n_{Rb,1}$ of [85]Rb and [87]Rb (including all spin states) in the source cavity is measured using absorption spectroscopy with device temperature varying between 330 K and 385 K. A representative spectrum measured in the source cavity at 363 K (Fig. 2a) shows a Doppler broadened spectrum consistent with thermal Rb vapor ($\bar{v} \approx 300$ m/s) and a density of $n_{Rb,1} \approx 2.4 \times 10^{18}$ m⁻³. The measured $n_{Rb,1}$ is consistent within experimental uncertainty with published values for the vapor pressure of liquid Rb metal across the temperature range probed.

The flux and angular divergence of the atomic beams are measured using fluorescence spectroscopy in the drift cavity. Fluorescence

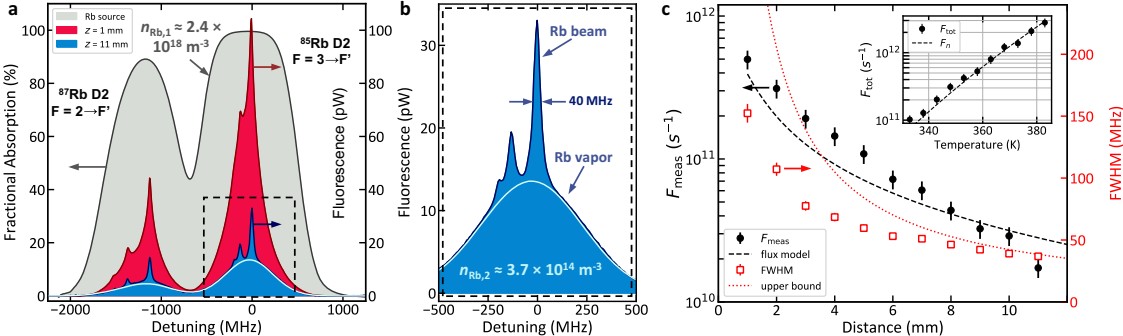

**Fig. 2 | Spectroscopic beam cell characterization. a** Source cavity absorption (gray) and drift cavity fluorescence at $z = 1\,mm$ (red) and $z = 11\,mm$ (blue) measures the Rb number density, flux, and the velocity distribution normal to the device surface at 363 K. **b** Fluorescence at z = 11 mm includes narrow peaks from the atomic beam signal as well as a broad signal corresponding to background Rb vapor (light blue curve). Passive and differential pumping generates a large (≈6500×) Rb partial pressure differential between the source and drift cavities. **c** The measured atomic beam flux $F_{meas}$ and spectral FWHM are plotted versus distance from the capillary array at 363 K. Flux prediction based on $n_{Rb,1}$ (black dashed lines) and the geometrical FWHM limit set by the fluorescence imaging (red dotted line) are shown. Inset shows the estimated total capillary flux $F_{tot}$ versus device temperature and comparison to the total expected capillary array flux $F_n$ (dashed line). Error bars represent 68% confidence intervals.

is collected using a 1:1 imaging system with ≈1.9% collection efficiency mounted at 45° from the beam axis in the *x-z* plane. The imaged area corresponds to a 1 mm × 1.4 mm region in the *x-y* plane. Fluorescence spectra scanning around the $^{85}$Rb F = 3 → F′ = 4 transition (labeled as zero optical detuning) are measured at varying distances along *z* from the capillary array. Example spectra at z = 1 mm and z = 11 mm at 363 K (shown in Fig. 2a, b) demonstrate narrow spectral features corresponding to the atomic beam signal and broader features arising from thermal background Rb vapor. The measured atomic beam flux is calculated from the number of detected atoms in the imaged volume $N_{det}$ (see methods) as $F_{meas} = N_{det} v_{beam}/L$, where $v_{beam}$ is the most probable longitudinal velocity of the atomic beam and *L* is the length over which the atoms interact with the probe beam. At 363 K and z = 1 mm, $F_{meas} = 5 \times 10^{11}\,s^{-1}$ and the FWHM of the fluorescence lines is ≈150 MHz, corresponding to a transverse velocity FWHM of ≈120 m/s. At this distance, ≈65% of the total capillary array is probed and the total atomic beam flux is estimated to be $F_{tot} = 7.7 \times 10^{11}\,s^{-1}$, consistent with the measured density in the source cavity and molecular flow predictions through the capillaries (see Fig. 2c inset).

Near the end of the drift cavity (z = 10 mm) $F_{meas} = 3.0 \times 10^{10}\,s^{-1}$ or ≈3.9% of the total flux due to the divergence of the atomic beam. This value matches the theoretical expectation (Fig. 2c dashed black line) of $3.2 \times 10^{10}\,s^{-1}$ based on $n_{Rb,1}$, the detected area of ≈1.4 mm², and the angular distribution function of our capillaries under molecular flow[21]. This agreement indicates that atomic beam loss due to collisions is consistent with zero within our level of systematic uncertainty. We note that the relatively strong divergence of this beam is typical of microcapillary collimation due to the presence of two atomic flux components, one that is direct (line-of-sight to the source) and the other indirect (diffuse scatter from capillary walls). For measurements of direct atoms ($\theta < a/L$), the atomic flux within this range is ≈$\sin^2(\theta)F_{tot}/w$ or ≈ 2.6% of $F_{tot}$ for the presented capillary geometry. The beam fluorescence FWHM is ≈40 MHz at this distance, set primarily by the range of *x*-velocities collected in the imaging system. At $F_{tot} = 7.7 \times 10^{11}\,s^{-1}$, 10 years of sustained operation would require 20 mm³ of metallic Rb. Reported flux and density values have an estimated statistical uncertainty of 15% and systematic uncertainty of 30%.

From the lack of measured collisional loss over a 10 mm distance in the drift cavity, we estimate an upper bound on the background pressure of ~1 Pa. For collisional loss to be significant given our systematic uncertainty in the absolute value of the atomic flux, the transport mean free path of Rb would need to be <1 cm, and this would require partial pressures of >12 Pa of $H_2$ or >3 Pa of $N_2$, which are common vacuum contaminants[23]. The true background pressure may

be significantly lower due to the high gettering efficiency of the NEGs for most common background gases including $H_2$, $N_2$, $O_2$, CO, and $CO_2$. The pumping speeds are estimated to be ≈1.4 L/s for $H_2$ and ≈0.14 L/s for CO at room temperature[24]. He is not pumped by the passive getters and the steady state He partial pressure will approach the ambient value of ≈0.5 Pa. However, this equilibration may be slowed by our use of low-He-permeation aluminosilicate glass[22]. Operation of the atomic beam device over many months indicates the rate of oxidation of deposited Rb is negligible.

Fluorescence from thermal background Rb vapor in the drift cavity is evident in all measured spectra, and the measured background Rb density is estimated as $n_{Rb,2} = 3.7 \times 10^{14}\,m^{-3}$ at $z \approx 11\,mm$ (see Fig. 2b), equivalent to a partial pressure of ≈ $2 \times 10^{-6}$ Pa. This density is ≈ 6500× lower than the Rb density in the source cavity due to differential pumping from the microcapillary array and passive pumping primarily from the graphite getters. Graphite getters are commonly used in atomic clocks due to graphite's affinity for intercalating alkali vapor, and the pressure differential implies a Rb pumping speed of ≈2 L/s for the ≈0.9 cm² of graphite surface area used. The graphite getters used (Entegris CZR-2) have high porosity and low strength relative to other commonly available graphites[25,26]. Recent work has demonstrated that graphite can also act as a solid-state reservoir for alkali-metal[27,28] and highly oriented pyrolytic graphite (HOPG) can serve both as an alkali getter and source depending on the operating temperature[29].

The beam device has been operated intermittently (≈1000 operation hours) over a period of 15 months without observed degradation of the vacuum environment. The deposited Rb metal in the source cavity is slowly consumed during normal device operation, and partial laser-thermal activation of the pill dispensers has been performed 9 times to deposit additional Rb metal in this cavity. Normal operation of the beam device is observed within minutes after activation and thermal equilibration of the device, and no period of excessive background pressure is observed. Complete activation of the pill dispensers in a single process is likely achievable but has not been attempted in this device. Saturation of the NEGs or graphite has not been observed, indicating that any real or virtual leaks are small, although absolute measurements of the pressure in the presented device have not been made.

The potential utility of the chip-scale atomic beam device is demonstrated using CPT Ramsey spectroscopy of the $^{87}$Rb ground state hyperfine splitting ($\nu_{HF} \approx 6.835\,GHz$) over a 10 mm distance, similar to previous laboratory CPT atomic beam clocks[15,30,31]. We address the $D_1$ F = 1,2 → F′ = 2 Λ-system (Fig. 3a) using two circularly

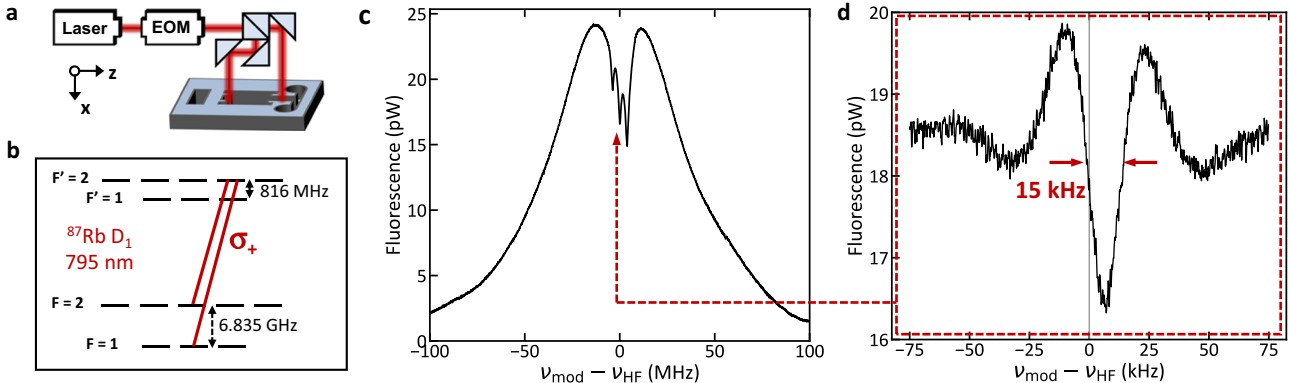

**Fig. 3 | Ramsey CPT spectroscopy of the atomic beam. a** Schematic of two-zone Ramsey interrogation of the atomic beam. **b** Level diagram illustrating the CPT Λ-systems. **c** Spectroscopic signal observed in 2nd Ramsey zone shows three, MHz-wide CPT features. **d** The central CPT feature contains the clock signal Ramsey fringe with ≈15 kHz width and ≈15% fluorescence contrast.

polarized, ≈250 μW laser beams propagating along the $x$-direction ($w_y \approx 550$ μm, $w_z \approx 150$ μm). The laser light is phase modulated at $\nu_{mod}$ using a fiber-based electro-optical modulator, and the carrier frequency and a 1st order sideband address the CPT Λ-system with approximately equal optical powers. The modulated light is split into two equal-length, parallel paths which intersect the atomic beam at $z = 1$ mm and $z = 11$ mm to perform two-zone Ramsey spectroscopy. Fluorescence from atoms in the 2nd zone (1.4 mm$^2$ imaged area) is collected on a Si photodiode with ≈1.9% efficiency. A magnetic field of ≈$2.8 \times 10^{-4}$ T is applied along the $x$-direction and separates the Zeeman-state dependent transitions.

CPT spectra measured in the second Ramsey zone at 363 K (Fig. 3c) demonstrate three, MHz-wide CPT resonances corresponding to $m_F = -1$, 0, and 1 Λ-systems from ≈1 μs long interaction with the optical fields in the 2nd Ramsey zone. At the center of each of these resonances (Rabi pedestals) are narrower Ramsey fringes arising from interaction with light in both Ramsey zones. The central Ramsey fringe (Fig. 3d) serves as our clock signal and has a fringe width of ≈15 kHz arising from the 30 μs transit time between the two zones. The signal height is ≈3.6 pW and the contrast relative to the one-photon fluorescence is ≈15%. Contrast is limited in our probing scheme by the spread of atoms among $m_F$ levels and optical pumping out of the Λ-system. Other probing schemes involving pumping with both circular polarizations can reduce loss outside of the desired $m_F$ level and increase fringe contrast[32–34]. The clock fringe is offset by ≈4.5 kHz from the vacuum value of $\nu_{HF}$ due to the second-order Zeeman effect. Optical path length uncertainty of 0.5 mm between the two Ramsey zones limits comparison to the vacuum value of $\nu_{HF}$ at the ≈400 Hz level.

An atomic beam clock is realized using the central Ramsey fringe to stabilize the CPT microwave modulation frequency. For this measurement, the beam device is heated to 392 K and the observed peak-to-valley height of the clock Ramsey fringe signal is ≈16 pW using 200 μW of optical power in each Ramsey zone. An error signal is formed using 150 Hz square wave modulation of the clock frequency at an amplitude of 11 kHz, and feedback is used to steer the microwave synthesizer's center frequency $\nu_{clock}$ with a bandwidth of ≈1 Hz. The synthesizer is referenced to a hydrogen maser, and a time series of $\nu_{clock}$ is recorded. The measured overlapping Allan deviation (ADEV) of the fractional frequency stability of $\nu_{clock}$ (Fig. 4) demonstrates a short-term stability of ≈$1.2 \times 10^{-9}/\sqrt{\tau}$ from 1 s to 250 s, limited by the signal height and the ≈13.5 fW/$\sqrt{Hz}$ noise equivalent power of the amplified Si detector used. Straightforward improvement in fluorescence collection efficiency and fringe contrast could improve the short-term stability below $1 \times 10^{-10}/\sqrt{\tau}$, similar to the performance of existing chip-scale atomic clocks[1]. Quantum projection noise limits the

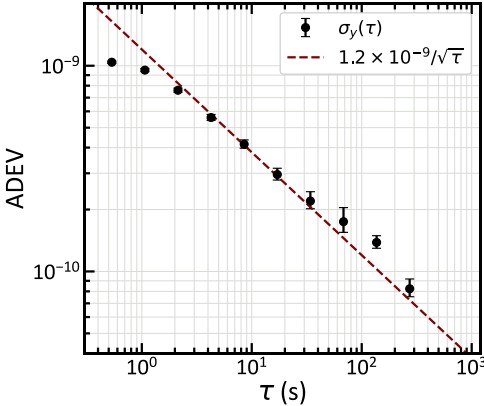

**Fig. 4 | Beam clock stability measurement.** The Allan deviation (ADEV) of the chip-scale atomic beam clock frequency (black points) is measured for integration times $\tau < 250$ s. The short-term fractional frequency stability $\sigma_y(\tau)$ is ≈$1.2 \times 10^{-9}/\sqrt{\tau}$ (red line) over this range. Error bars represent 68% confidence intervals.

potential stability of the presented measurement to ≈$9 \times 10^{-12}/\sqrt{\tau}$, assuming a thermal $^{87}$Rb beam at 392 K, a detectable flux of $1.8 \times 10^{10}$ s$^{-1}$, and a fringe contrast of 25%.

## Discussion

We have demonstrated a chip-scale atomic clock based on miniaturized atomic beams. The key components of the passively pumped atomic beam device are planar, lithographically defined structures etched in Si and glass wafers, compatible with volume microfabrication. The 10-channel microcapillary array etched into one of the Si device layers provides a total atomic flux of ≈$7.7 \times 10^{11}$ s$^{-1}$ at 363 K, and ≈3.9% of the atoms pass through a 1.4 mm$^2$ detection area 10 mm downstream. The measured performance of the atomic beam matches expectations based on molecular flow through the collimator array with no free parameters, indicating that collisions with background gases are minimal and the background pressure is ~1 Pa or lower. Passive and differential pumping of the Rb vapor supports the ≈6500× Rb partial pressure differential between the source and drift cavities and enables high beam flux while minimizing the background Rb pressure in the drift cavity. The presented beam system has been operated intermittently for 15 months without degradation of the vacuum environment or saturation of the passive pumps.

The realization of a microwave Ramsey CPT beam clock demonstrates the potential utility of the atomic beam device. CPT Ramsey fringes are measured using the atomic beam over a 10 mm distance, demonstrating atomic coherence across the drift cavity. A clock signal

is formed using the magnetically insensitive, $m_F = 0$ transitions between the $^{87}$Rb hyperfine ground states, and Ramsey fringes at an operating temperature of 392 K are measured to be 15 kHz-wide with 16 pW of CPT signal. This clock signal is used to stabilize the microwave oscillator driving the CPT transitions, and the clock demonstrates a short-term fractional frequency stability of $\approx 1.2 \times 10^{-9}/\sqrt{\tau}$ from 1 s to 250 s. The presented short-term stability is limited by the available signal-to-noise ratio (SNR) of $\approx 1200$ at 1 s, and improvement of the short-term stability below $1 \times 10^{-10}/\sqrt{\tau}$ appears feasible, competitive with existing buffer gas-based miniature atomic clocks, by straightforward improvement to the collection optics and use of a higher contrast pumping scheme[32,33,35].

The presented beam clock approach has the potential to exceed existing chip-scale atomic clocks in both long-term stability and accuracy[36]. Commercial atomic beam clocks based on microwave excitation of the clock transition using a Ramsey length of $\approx 15$ cm achieve a stability at 5 days of $10^{-14}$ and an accuracy of $5 \times 10^{-13}$. Many of the key systematics in beam clocks scale with the clock transition linewidth, and hence inversely with the Ramsey distance, implying that a 15 mm beam clock could achieve stability at the $10^{-13}$ level. Work on CPT atomic beam clocks using Na and a 15 cm Ramsey length achieved a stability of $1.5 \times 10^{-11}$ at 1000 s without evidence of a flicker floor[31,37]. Projected to Cs with a Ramsey length of 15 mm implies an achievable stability at the level of $1.0 \times 10^{-11}$ at 1000 s, equivalent to less than 100 ns timing error at 1 day of integration. Realization of this stability will depend on managing drift in the optical, vacuum, and atomic environments in a fully miniaturized beam clock system.

The leading systematic shifts that will impact a compact beam clock include Doppler shifts, Zeeman shifts, end-to-end cavity phase shifts, collisional shifts, and light shifts. Each of these shifts has been studied extensively in conventional microwave atomic beam frequency standards[13,38] and in CPT beam clocks[31]. We evaluate these requirements assuming a stability goal of $10^{-12}$, equivalent to $\approx 6.8$ mHz stability of $\nu_{clock}$, for a 1.5 cm Ramsey length. Optical path length instability of the CPT laser beam can lead to both Doppler shifts and end-to-end cavity phase shifts. Doppler shifts arise from CPT laser beam pointing drift (thermal or aging) along the atomic beam axis and shifts the clock frequency at $\approx 7.5$ kHz rad$^{-1}$, requiring μrad beam pointing stability to reach $10^{-12}$ frequency stability. Optical path length stability at the 10 nm level is needed to minimize end-to-end cavity phase shifts. This shift can be minimized using symmetrical Ramsey beam paths, which make thermal expansion common mode along the two Ramsey arms and largely eliminates the bias. Asymmetrical path length variation can arise from thermal gradients along the beam paths and will induce clock shifts. For 15 mm beam paths fabricated using glass or Si substrates, 100 mK temperature uniformity is sufficient to achieve the desired stability.

Collisional shifts place limits on the vacuum stability required in the atomic beam device. Common background gases such as $H_2$ and He induce collisional shifts of $\nu_{HF}$ at the level of 5 Hz Pa$^{-1}$, and 1 mPa pressure stability is needed to achieve $10^{-12}$ fractional frequency stability[23,39]. Given the inferred pressure of ~1 Pa in our device at 363 K, 100 mK temperature stability is sufficient assuming zero background pressure variation. Stabilizing the He partial pressure in passively pumped devices is challenging due to the high He diffusivity in many materials and insufficient getter material for He. We use low-He-permeation aluminosilicate glass to reduce the rate of He ingress, stabilizing He partial pressure variations[22,40–42]. Collisions with background Rb atoms along the drift cavity generate spin-exchange shifts of $\nu_{HF}$, and the magnitude of the Rb-Rb collisional shift depends on the occupancy ratio between ground state hyperfine levels before CPT interrogation. Assuming optical pumping into the F = 2 ground state, $\nu_{HF}$ shifts at $\approx 3400$ Hz Pa$^{-1}$. The total clock shift is estimated to be $\approx 6.8$ mHz for our demonstrated $\approx 2 \times 10^{-6}$ Pa Rb background partial pressure[43] and places lax requirements on the background pressure

stability. Intra-beam collisional shifts also exist at approximately the same level and can be reduced using cascaded collimators to reduce the atomic beam density[18].

Light shifts are another significant source of clock instability and can arise from both the ac-Stark effect[44] as well as incomplete optical pumping into the CPT dark state[45,46]. The magnitude and sign of the light shifts can depend strongly on the intensity ratio of the CPT driving fields and the pumping scheme used[47], and the shift scales inversely with the Ramsey time. We estimate the light shift sensitivity in the proposed geometry at the level of $1 \times 10^{-12}$ for a 0.1% change in the CPT field intensity ratio based on measured light shifts in a cold-atom clock using $\sigma_+/\sigma_-$ optical pumping, nominally equal CPT field intensities, and a 4 ms Ramsey time[47]. This level of stability may require use of active monitoring of the optical modulation used to generate the CPT fields[42,48,49]. Several methods have been developed to manage light shifts in atomic clocks using multiple measurements of the clock frequency, such as auto-balanced Ramsey spectroscopy[50,51] or power-modulation spectroscopy[52,53]. Zeeman shifts arise from variations in the quantization magnetic field, and at a field of $\approx 10^{-4}$ T (sufficient to separate the magnetically sensitive $m_F \neq 0$ transitions from the clock transition), the Zeeman effect shifts $\nu_{HF}$ by $\approx 575$ Hz. This dictates a field stability at six parts-per-million (ppm), which can be achieved using intermittent interrogation of a $m_F \neq 0$ transition to correct the field strength.

Considering each of the common sources of drift summarized in Table 1, an ultimate fractional frequency stability at or below the level of $10^{-12}$ appears feasible in a chip-scale atomic beam clock. The presented beam clock presents a path for realizing low size, weight, and power (low-SWaP) atomic clocks. Future efforts will focus on realizing this long-term clock stability goal using integration with micro-optical and thermal packaging to produce a fully integrated device at the size- and power-scale of existing CSACs. Such a device should achieve sub-μs timing error at a week of integration and would contribute to low-SWaP timing holdover applications. The chip-scale beam device presented here is a general platform for quantum sensing, and future work using this system could explore applications including inertial sensing using atom interferometry, electrometry using Rydberg spectroscopy, and higher-performance compact clocks using optical transitions.

## Methods
### Fabrication of atom beam device
Features including two internal cavities and the microcapillary array are etched into Si wafers using DRIE. The beam cell is assembled by first anodically bonding the Si layers, intermediate glass layer, and one encapsulating glass layer to create a preform structure. The preform bonds are performed in air at 623–673 K using a bonding voltage of 800–900 V for several hours. The Rb pills, graphite rods, and NEGs are loosely held in cavities etched in the 2-mm-thick Si wafer, leaving room for expansion during operation. The final bond, which hermetically seals the package, occurs in a UHV chamber using an electrode to press the final encapsulating glass layer onto the preform. The device is vacuum baked at $\approx 1 \times 10^{-5}$ Pa and 520 K for 20 h to reduce volatile contaminants, and the NEGs are thermally activated using a few W of

**Table 1 | Expected chip-scale clock systematics**

| Clock systematic shift | Sensitivity (δf/f) | Control, $10^{-12}$ stability |
|---|---|---|
| Doppler | $1 \times 10^{-6}$ rad$^{-1}$ | $\delta\theta < 1 \times 10^{-6}$ rad |
| Zeeman | $2 \times 10^{-3}$ T$^{-1}$ | $\delta B < 0.6$ nT |
| Collisional ($H_2$, $N_2$) | $7 \times 10^{-10}$ Pa$^{-1}$ | $\delta T < 0.1$ K |
| Collisional (Rb) | $5 \times 10^{-7}$ Pa$^{-1}$ | $\delta T < 100$ K |
| Light ($R = P_1/P_2$) | $1 \times 10^{-9}$ R$^{-1}$ | $\delta R < 1 \times 10^{-3}$ |
| Cavity end-to-end | $1 \times 10^{-6}$ cm$^{-1}$ | $\delta T < 0.1$ K |

Systematics are calculated for a Rb beam clock with 15 mm Ramsey length, and requirements for experimental control are estimated to achieve $10^{-12}$ fractional frequency stability.

975 nm laser light focused onto each getter for ≈600 s. The NEGs are observed to glow a red color during activation, consistent with a temperature ~1170 K. After NEG activation, the final anodic bond hermetically seals the device. Pressure in the UHV chamber is maintained at ≈3.5 × 10$^{-5}$ Pa for the bond with the cell at ≈640 K and bonding voltage of −1800 V for 21 h. The sealed device is removed from the UHV bonding chamber and mounted on a temperature-controlled plate for operation in air. Pill dispensers (1 mm diameter, 0.6 mm thickness, 0.4 mg total Rb content) are activated using laser heating with a few W of 975 nm light to a temperature of ≈950 K until a stable Rb density is observed in the source cavity.

**Atom beam flux characterization**

The atomic beam flux is measured using fluorescence spectroscopy on the Rb D$_2$ transitions as shown in Fig. 2[19,54]. Spectroscopy is performed using a 5 μW elliptical laser beam traveling normal to the beam device surface with 1/e$^2$ radii of $w_y ≈ 2100$ μm and $w_z ≈ 350$ μm. The peak intensity is ≈0.1 mW/cm$^2$, and the low intensity limits optical pumping during transit through the probing beam. Atomic fluorescence is imaged using a 1:1 imaging system mounted at 45 degrees in the x-z plane to collect ≈1.9% of the atomic fluorescence onto a 1 mm × 1 mm Si photodiode. The effective probed volume at constant interrogating intensity is ≈1 mm × 1.4 mm × $w_z\sqrt{\pi/2}$ or ≈0.6 mm$^3$, and this volume can be translated across the drift region. The factor $L_z = w_z\sqrt{\pi/2}$ accounts for the Gaussian intensity variation along the z-direction. The flux through this volume $F_{meas} = N_{meas}v_{beam}/L_z$ where $N_{meas}$ is the number of atoms measured in the probed volume, $v_{beam} = \sqrt{3k_BT/m}$ is the most probable longitudinal velocity, k$_B$ is Boltzmann's constant, and $m$ is the atomic mass. We measure $N_{det}$ using the integrated spectrum of radiated power $\Phi_{Total} = \int P(\omega)d\omega$ across the $^{85}$Rb $F = 3 −> F' = 4$ transition where $P(\omega)$ is the measured optical power at angular detuning ω. At low saturation intensity the integrated spectral power per atom $\Phi_0 = \frac{hc\pi}{4\lambda}s\Gamma^2$ and $N_{meas} = \frac{\Phi_{Total}}{\Phi_0}$, where $h$ is Plank's constant, $c$ is the speed of light, $\lambda$ is the D$_2$ transition wavelength, $s$ is the transition saturation parameter, and $\Gamma ≈ 2\pi × 6.067$ MHz is the natural linewidth of the D$_2$ transition.

## Data availability

Data underlying the results of this study are available from the authors upon request.

## Code availability

The codes that support the findings of this study are available from the authors upon request.

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

## Acknowledgements

We acknowledge Andrew Ludlow and Ying-Ju Wang for their comments on this document. We acknowledge Elizabeth Donley and Azure Hansen for contributions to device design; Juniper Pollock for helpful discussions; and Susan Schima, Daron Westly, and Durga Gajula for support in device fabrication. Official contribution of the National Institute of Standards and Technology; not subject to copyright in the United States. Mention of commercial products is for information only and does not imply recommendation or endorsement by NIST. This work was supported by the NIST on a Chip program (G.D.M., A.S., J.K., and W.R.M.). Award 70NANB18H006 from U.S. Department of Commerce, NIST (G.D.M., A.S.), and award No. N00014-20-1-2429 from the Office of Naval Research (C.L., C.R.). A patent is pending (U.S. Patent Application No. 63/302,308).

## Author contributions

C.L., G.D.M., C.R., and J.K. contributed to the atomic beam device design. G.D.M., C.L., J.K., C.R., and W.R.M. contributed to the beam device development and fabrication. G.D.M., A.S., and W.R.M. performed measurements and data analysis of the atomic beam device and the atomic beam clock. All authors contributed to the CPT Ramsey protocol, interpretation of results, and writing of this manuscript.

## Competing interests

The authors declare no competing interests.
