## [Peer Review File · Nature Communications]

A chip-scale atomic beam clockREVIEWERS' COMMENTS

Reviewer #1 (Remarks to the Author):

In their manuscript, Martinez et al., report on the realization of a chip-scale atomic beam clock. Based on previous work, the heart of it is formed by a microfabricated microchannel array which in this work is integrated into a full-blown, wafer level vacuum cell including passive pumps and a source & interrogation section. With a current performance of $1e-9/\sqrt{\tau}$ limited by detection noise, the authors perform a detailed discussion of routes towards competition and even outperformance of existing, low SWaP-budget chip-scale clocks regarding both stability and accuracy.

I congratulate the authors to their results which I deem an exciting and fascinating step towards highly integrated quantum technology platforms based on cold atoms. The obtained results are presented clearly and it is a pleasure to read the paper which keeps an interesting balance between the big picture, technical details, and careful data analysis. The results will draw a lot of attention in the quantum technology community and I fully second the view that this will lead to applications far beyond frequency standards.

I consider the work very appropriate for Nat. Commun. and recommend to accept the manuscript without revisions.

The only point which is definitely in need of editing my be:

I 146: I believe the authors wanted to refer to Fig. 2c instead of 3c.

Reviewer #2 (Remarks to the Author):

This paper describes progress toward a much-needed highly compact, long-term frequency/time reference. Using an atomic beam avoids wall and pressure shifts which should give it superior long-term stability compared to vapors.

The results show sufficiently encouraging short-term stability that the improvements for future long-term stability can now begin. The work plan for this is outlined in detail in the paper.

I have only optional revision:

Comparing achieved stability, projected stability, commercial atomic beam, and vapor clock stability is hard to follow. Suggest making a table including at least short-term, mid-term (1000 s) and long-term entries. Similar to the table for systematic shifts.

The lifetime of the vacuum system as measured so far is encouraging. At some point the need for frequent thermal laser activation will have to be addressed in more detail, especially for applications involving remote deployment. Although the authors already briefly discuss some options, it would be good to see it in the context of potential applications.

Reviewer #3 (Remarks to the Author):

The size of the atomic clock is intriguing and will interest many readers, already the comparison with the peanut is pretty neat. The data are concise, and consistent. This is valid research in science and technology development at the front of the state of the art in miniaturization of such devices.

The work is scientifically sound and technologically interesting. The data provided are sufficiently detailed to provide a very clear picture of the working principles and capabilities. It is easy to read.

Some of the technologies developed here may also serve other parts of the cold atom community. The miniaturized vacuum, atom beam, ...The durability of >15 months without degradation is already a good indication that the system has high technology readiness.

In Figure 4 there is a beginning of a deviation in ADEV from the dashed line. It would have been nice to show a "1000 s point", to see the actual limit, but the argument of the paper is also made as it is.

The authors emphasize the CPT feature, assuming that everyone knows what coherent population trapping is. I guess this should be spelled out early on, to provide accessibility to more readers.

The references appear fair and balanced.

Even though the accuracy of the clock is many orders of magnitude lower than the state of the art in optical clocks, and a sugar cube scale clock had been published before, I think the technological finesse warrants publication in Nature Comm. There are very many scientific and practical applications of tiny clocks with a precision of 9 digits per second and prospects of getting as good as the best 19inch rack RF atomic clocks with a much smaller form factor.

I recommend the paper for publication as is (modulo very few typos).

The authors thank the Reviewers for their comments on the manuscript. We provide a detailed response to their suggestions below in blue.

Regards,

William McGehee

5/3/2023

REVIEWERS' COMMENTS

Reviewer #1 (Remarks to the Author):

In their manuscript, Martinez et al., report on the realization of a chip-scale atomic beam clock. Based on previous work, the heart of it is formed by a microfabricated microchannel array which in this work is integrated into a full-blown, wafer level vacuum cell including passive pumps and a source & interrogation section. With a current performance of $1e-9/\sqrt{\tau}$ limited by detection noise, the authors perform a detailed discussion of routes towards competition and even outperformance of existing, low SWaP-budget chip-scale clocks regarding both stability and accuracy.

I congratulate the authors to their results which I deem an exciting and fascinating step towards highly integrated quantum technology platforms based on cold atoms. The obtained results are presented clearly and it is a pleasure to read the paper which keeps an interesting balance between the big picture, technical details, and careful data analysis. The results will draw a lot of attention in the quantum technology community and I fully second the view that this will lead to applications far beyond frequency standards.

I consider the work very appropriate for Nat. Commun. and recommend to accept the manuscript without revisions.

We thank the reviewer for their positive view of our work and their suggestion to accept without revision

The only point which is definitely in need of editing my be:

I 146: I believe the authors wanted to refer to Fig. 2c instead of 3c.

We apologize for the confusion. We have corrected this mistake.

Reviewer #2 (Remarks to the Author):

This paper describes progress toward a much-needed highly compact, long-term frequency/time reference. Using an atomic beam avoids wall and pressure shifts which should give it superior long-term stability compared to vapors.

The results show sufficiently encouraging short-term stability that the improvements for future long-term stability can now begin. The work plan for this is outlined in detail in the paper.

I have only optional revision:

Comparing achieved stability, projected stability, commercial atomic beam, and vapor clock stability is hard to follow. Suggest making a table including at least short-term, mid-term (1000 s) and long-term entries. Similar to the table for systematic shifts.

We apologize for the confusion. There is a recent publication from B. Schmittberger Marlow at MITRE which reviews existing and emerging frequency standards in great details. We have added a reference to this publication to aid the reader in a more complete fashion than we feel a table would provide.

The lifetime of the vacuum system as measured so far is encouraging. At some point the need for frequent thermal laser activation will have to be addressed in more detail, especially for applications involving remote deployment. Although the authors already briefly discuss some options, it would be good to see it in the context of potential applications.

We thank the author for this comment. We have indicated in the text that full activation of the Rb source is likely possible. We have not attempted this yet as we are early in the development of this technology, and any further statement would be beyond the scope of this work.

Reviewer #3 (Remarks to the Author):

The size of the atomic clock is intriguing and will interest many readers, already the comparison with the peanut is pretty neat. The data are concise, and consistent. This is valid research in science and technology development at the front of the state of the art in miniaturization of such devices.

The work is scientifically sound and technologically interesting. The data provided are sufficiently detailed to provide a very clear picture of the working principles and capabilities. It is easy to read.

Some of the technologies developed here may also serve other parts of the cold atom community. The miniaturized vacuum, atom beam, ...The durability of >15 months without degradation is already a good indication that the system has high technology readiness.

We thank the reviewer for their feedback and agree that this vacuum technology will eventually impact the compact cold atom community.

In Figure 4 there is a beginning of a deviation in ADEV from the dashed line. It would have been nice to show a "1000 s point", to see the actual limit, but the argument of the paper is also made as it is.

We are currently studying the long-term stability of our clock system, and these results will be presented in subsequent publication. Our existing test apparatus was built for preliminary device characterization, and beyond a 1000 s we believe are limited by this test apparatus and not the beam clock device. Our hope is to probe well beyond 1000 s using our new testing apparatus.

The authors emphasize the CPT feature, assuming that everyone knows what coherent population trapping is. I guess this should be spelled out early on, to provide accessibility to more readers.

We thank the reviewer for suggesting this improvement. We have added details to the introduction with better introduce CPT and have reference the review paper by J. Vanier on CPT based atomic clocks for those interested in greater detail.

The references appear fair and balanced.

Even though the accuracy of the clock is many orders of magnitude lower than the state of the art in optical clocks, and a sugar cube scale clock had been published before, I think the technological finesse warrants publication in Nature Comm. There are very many scientific and practical applications of tiny clocks with a precision of 9 digits per second and prospects of getting as good as the best 19inch rack RF atomic clocks with a much smaller form factor.

I recommend the paper for publication as is (modulo very few typos).

We thank the reviewer for their positive comments and have corrected the typos.